# PhysE-Inv: A Physics-Encoded Inverse Modeling Approach for Arctic Snow Depth Prediction

## Abstract

The accurate estimation of Arctic snow depth ($h_s$) remains a critical time-varying inverse problem due to the extreme scarcity and noise inherent in associated sea ice parameters. Existing process-based and data-driven models are either highly sensitive to sparse data or lack the physical interpretability required for climate-critical applications. To address this gap, we introduce PhysE-Inv, a novel framework that integrates a sophisticated sequential architecture, an LSTM Encoder-Decoder with Multi-head Attention and physics-guided contrastive learning, with physics-guided inference. Our core innovation lies in a surjective, physics-constrained inversion methodology. This methodology first leverages the hydrostatic balance forward model as a target-formulation proxy, enabling effective learning in the absence of direct $h_s$ ground truth; second, it uses reconstruction physics regularization over a latent space to dynamically discover hidden physical parameters from noisy, incomplete time-series input. Evaluated against state-of-the-art baselines, PhysE-Inv significantly improves prediction performance, reducing error by 20% while demonstrating superior physical consistency and resilience to data sparsity compared to empirical methods. This approach pioneers a path for noise-tolerant, interpretable inverse modeling, with wide applicability in geospatial and cryospheric domains.

## 1 Introduction

Snow depth ($h_s$) exerts a first-order control on Arctic sea ice thickness, yet reliable observations of it remain remarkably scarce. For instance, widely used reanalysis products such as ERA5 do not provide direct measurements of snow depth over sea ice (Hersbach et al., 2020), despite its well-established influence on sea ice thermodynamics (Li et al., 2024). Compounding this data gap, the most available driving observations, such as snow density ($\rho_s$), are inherently noisy and sparse due to measurement complexities. To address this gap, we exploit Physics-encoded Neural Networks (PeNNs) (Faroughi et al., 2022), a class of models widely applied in scientific analysis to implement inverse modeling. However, a standard PeNN approach is insufficient to overcome the core issue of extreme data scarcity and input noise. Therefore, our work introduces a novel framework that not only encodes sea ice physics derived from the hydrostatic balance equation (Kwasniok, 2022) but, critically, leverages this equation as a target-formulation proxy to enable the estimation of hidden parameters and yield physically consistent prediction of snow depth ($h_s$) from the noisy observational data.

Inverse modeling is a powerful tool for inferring intrinsic physical parameters and uncovering hidden characteristics of complex physical phenomena. These models extract meaningful insights from systems in various fields, including lake temperature modeling (Tayal et al., 2022), tomography (Bubba et al., 2019), seismic waveform analysis (Sun et al., 2020), materials science (Liao & Li, 2020), and hydrology (Ghorbanidehno et al., 2020). However, a common limitation is that many of

these methods rely on physical priors or posterior distributions for parameter estimation (Karumuri & Bilionis, 2024). For example, the accuracy of the predicted variable directly depends on the accuracy of the physical mapping from other variables. This dependency can be problematic because any errors in the initial mapping will propagate through the model during training, potentially affecting the overall performance. Furthermore, in data-sparse domains like the Arctic, the forward model is highly sensitive to noisy input, exacerbating this error propagation. While an end-to-end learning approach can offer a streamlined solution, it can also limit the ability to embed additional physical constraints that are relevant to the system (Faroughi et al., 2022). This is because the end-to-end model is highly dependent on its initialization process and the pre-defined forward mapping.

Furthermore, inferring parameters through inverse modeling using complex forward-inverse flow transformations (Tarantola, 2005; Ghosh et al., 2022; Tayal et al., 2022) can be a computationally demanding process. These works often rely on a bijective mapping between the input and latent spaces. While this one-to-one correspondence is suitable for static systems (e.g., unchanging material properties), its strict nature is restrictive in the context of a dynamic physical system like Arctic sea ice, where underlying parameters are time-varying. A bijective mapping may not be flexible enough to capture these evolving relationships, as it rigidly links a single input state to a single output state, limiting its ability to account for the continuous evolution of the system. Additionally, these methods build upon traditional self-supervised learning that extracts general statistical patterns from unlabeled data. However, the learned features, which are not constrained by physical principles, might not be physically meaningful or suitable for parameter estimation in a specific domain like snow depth prediction. This points to a clear need for a physics-guided contrastive learning approach that embeds domain knowledge directly into the feature-learning process.

To overcome the limitations of both traditional inverse modeling and conventional self-supervised learning, we introduce a novel, simplified hybrid framework called PhysE-Inv (Physics-Encoded Inverse modeling). This framework utilizes a surjective inversion mapping for time-varying parameter estimation, enabling it to exploit all possible values the predicted variable could take. It operates on the principle that the temporal evolution of a system can be effectively modeled using linear dynamics present in its physical variables (Kwasniok, 2022). We embed a linear physical equation derived from the hydrostatic balance Kwok & Cunningham (2008) to relate the cause-and-effect phenomena between sea ice and snow variables, leveraging the physical equation as a target formulation proxy for inversion. The learning mechanism within PhysE-Inv specifically addresses the pitfalls of generic self-supervised learning by integrating a physics-guided contrastive learning approach via physics encoding. This is critical because it compels the model's latent space to capture the physically meaningful relationships between Arctic snow and sea ice variables. By enforcing these relationships through contrastive loss, we ensure the latent space is optimally structured for the subsequent inversion and estimation of hidden parameters needed for physically consistent predictions of snow depth.

The main contributions of this paper are summarized as follows: **(1)** We present PhysE-Inv, a novel hybrid framework that leverages a surjective inversion mapping integrated into a physics-guided contrastive learning process. The novelty of this work lies not in the creation of a novel network architecture, but in the constrained inversion methodology, which enables a more direct and efficient optimization for predicting snow depth through inverse modeling. **(2)** Moving beyond traditional inverse modeling, our framework achieves effective physics encoding by embedding constraints from the hydrostatic balance equation of Arctic sea ice thickness directly into the neural network. This ensures physically consistent inferences, leading to more accurate snow depth predictions. **(3)** We demonstrate the effectiveness of our proposed model by comparing it with multiple baselines, showing superior performance and resilience to data sparsity.

## 2 RELATED WORKS

### 2.1 PHYSICS-GUIDED MACHINE LEARNING

Physics-encoded Neural Networks (PeNNs) represent powerful approach for integrating scientific principles into machine learning Faroughi et al. (2022); Willard et al. (2022); Karpatne et al. (2024). By incorporating physical laws in the form of differential and linear equations Rao et al. (2021); Chen et al. (2018); Kovachki et al. (2021); Innes et al. (2019), these models not only show improved performance but also adhere more closely to physical laws, transforming traditional black-box algorithms into more interpretable models. Building on the rigor introduced by PeNNs, machine learning–enhanced inverse modeling offers a powerful way to uncover hidden physical parameters that govern observable geospatial phenomena, many of which cannot be measured directly. This approach has been widely applied in fields such as hydrology Ghosh et al. (2022), water flow studies Mo et al. (2020), and lake temperature modeling Tayal et al. (2022). In seismic waveform inversion, for example, researchers have begun integrating theoretical knowledge of seismic wave propagation into deep learning frameworks Adler et al. (2021). Deep neural networks are also being employed to tackle electrical impedance tomography (EIT) problems, which involve inverting the highly nonlinear and high-dimensional Dirichlet-to-Neumann (DtN) map Fan & Ying (2020). In addition, innovative architectures such as SwitchNet have been developed to address forward and inverse scattering problems for wave equations, offering computational efficiency while capturing the global nature of scattering phenomena Khoo & Ying (2021).

Most physics-informed machine learning (PIML) approaches, such as Physics-Informed Neural Networks (PINNs) Raissi et al. (2019), are designed for learning or solving nonlinear partial differential equations (PDEs), ordinary differential equations (ODEs), or complex dynamical and turbulent processes Nguyen et al. (2025); Lu et al. (2021). These frameworks are powerful for numerical simulation, system identification, and discovering new dynamical laws, but they are not well suited to the simple, linear inverse modeling setting considered in this work, where the goal is to infer hidden physical parameters from incomplete real-world observations. Our task focuses on estimating unobserved but physically meaningful quantities rather than solving a forward dynamical system, making the assumptions and machinery of PDE-based PIML unnecessarily heavy and often incompatible with the available data. Similarly, foundation models such as ClimaX Nguyen et al. (2023) are optimized for large-scale climate prediction and cross-variable representation learning under conditions where abundant training data exist. Their objectives and data regimes differ substantially from ours: we target parameter estimation in data-scarce environments, where the structure of the inverse map is weakly constrained and the physical signal must be recovered from limited and partially observed inputs. As a result, our surjective inverse estimation task lies outside the problem class these foundation models are designed to address.

### 2.2 SELF-SUPERVISED LEARNING

While effective, ML-enhanced inverse methods often require extensive labeled data. To address this data scarcity challenge, the self-supervised paradigm has also proven effective in addressing inverse problems across a wide range of scientific disciplines, extending beyond traditional domains like natural language processing Fang et al. (2020) and computer vision Bardes et al. (2022) to include scientific modeling Scotti et al. (2023) and noninvasive medical digital twins Kuang et al. (2025). By exploiting complex pretext tasks and multi-stage training, these methods can extract meaningful representations directly from underlying physical structures Bardes et al. (2022), which in turn enables parameter estimation without the need for explicit labels Liu et al. (2021); Jing & Tian (2020). A limitation of these methods is their heavy reliance on generative and regularization losses, which necessitate extensive hyperparameter tuning. Moreover, the use of bijective mappings (e.g., Ghosh et al. (2022)) assumes perfect invertibility, which is not plausible with real-world data. Despite these advances, a research gap remains. Our work proposes a unified framework that merges the rigor of physics-encoded methods (using known laws with unknown variables) with direct optimization to solve time-varying inverse problems for unobserved data.

## 3 DATA AND METHODOLOGY

### 3.1 DATASET

The dataset comprises ERA5 reanalysis data Hersbach et al. (2020) from the European Centre for Medium-Range Weather Forecasts (ECMWF). ERA5 ingests a wide variety of observational data, a significant portion of which comes from remote sensing instruments Hersbach et al. (2020). We acquired spatiotemporal data from January 1, 1995, to 2024 (10,958 time steps), specifically for the central Arctic Ocean region enclosed by the highlighted orange boundary in Figure 4, which roughly corresponds to the latitude range of approximately 70°N to 85°N. The data has a spatial resolution of $0.25° \times 0.25°$ (approximately 25 km). It includes key parameters related to snow depth and sea ice thickness: snow albedo, snow density, and sea ice concentration. We therefore process these variables using a proxy model derived from the hydrostatic balance equation to generate target proxy data that is eventually used in the inversion process to estimate hidden characteristics.

### 3.2 PROBLEM FORMULATION AND PRELIMINARIES

The observational data is first defined. The primary input time series is given as $\mathbf{X} = [x_1, x_2, \ldots, x_T]$, where $x_t$ represents the measurement at time step $t$, and $T$ denotes the total number of time steps. A corresponding augmented input sequence (Ref, Fig 1), $\mathbf{X}' = [x'_1, x'_2, \ldots, x'_T]$, is generated to facilitate contrastive learning. The target variable is $\mathbf{Y} = [y_1, y_2, \ldots, y_T]$, where $y_t$ is the observed output at time step $t$.

We aim to model the relationship between the input $\mathbf{X} = \rho_s$ and the target $\mathbf{Y} = h_i$ derived via the proxy model, particularly in the context of physics-constrained snow depth prediction, by integrating snow and sea ice properties. To achieve this, we adapt the hydrostatic balance equation, which, as stated by Kwok & Cunningham (2008), describes the equilibrium where the weight of ice and snow is balanced by the buoyant force of seawater (equation 2). Using this relation, we generate proxy $\mathbf{Y}$ labels to inject physical attributes into the training process.

## 4 INVERSE PROBLEM: PREDICTING $h_s$

In this study, we frame the inverse problem as predicting the hidden physical parameter $h_s$ using known geophysical observations such as sea ice concentration, snow albedo, and density fields. The difficulty arises because computing $h_s$ analytically requires inputs such as ice thickness ($h_i$), freeboard ($f_b$), and snow density ($\rho_s$), which are actually unobserved in ERA5.

$$g(\mathbf{x}) = h_s \approx \mathbf{F}^{-1}(h_i, \rho_s) \tag{1}$$

where $g$ denotes the learned estimator that recovers snow depth from the available observations.

This problem is fundamentally ill-posed because a unique one-to-one inversion between the observed input vector $\mathbf{X}$ and the target $\mathbf{Y}$ is impossible. For instance, several distinct combinations of ice thickness and snow density can result in the same observed snow depth. To address this inherent non-uniqueness and to stabilize the solution, we model the inverse relationship as a surjective mapping. This assumption is necessary because the large, high-dimensional input space ($\mathbf{X}$) must be mapped onto a much smaller, physics-constrained output space ($h_s$), meaning the input space can be mapped onto multiple potential outputs. Our framework uses this surjective mapping assumption to estimate the hidden physical parameters ($\alpha, \beta, \gamma$) at every time step $t$, which are used to reconstruct a physics-constrained prediction $h_s$.

## 4.1 PROPOSED PROXY MODEL

The selection of a stable ground truth for training is essential. Since direct observation of key parameters is uncertain in ERA5, we define our target using an analytically derived ice thickness proxy model based on the principle of hydrostatic equilibrium.

This derivation establishes the core thickness equation, which serves as the starting point for any proxy model.

The total weight of the snow and ice column equals the buoyant force (the weight of the displaced water):

$$\rho_i h_i + \rho_s h_s = \rho_w h_{\text{sub}} \tag{2}$$

The total depth of the column is the sum of the submerged depth ($h_{\text{sub}}$) and the surface elevation (freeboard, $f_b$):

$$h_i + h_s = h_{\text{sub}} + f_b$$

Rearranging to isolate the submerged depth:

$$h_{\text{sub}} = h_i + h_s - f_b \tag{3}$$

Substitute Equation (3) back into the Hydrostatic Balance (2):

$$\rho_i h_i + \rho_s h_s = \rho_w(h_i + h_s - f_b)$$

**Solving for $h_i$:** Expand, group all terms containing $h_i$ on one side, and factor:

$$\rho_i h_i + \rho_s h_s = \rho_w h_i + \rho_w h_s - \rho_w f_b$$

$$h_i(\rho_i - \rho_w) = h_s(\rho_w - \rho_s) - \rho_w f_b$$

Finally, isolating $h_i$ yields the standard forward equation:

$$h_i = \frac{h_s(\rho_w - \rho_s) - \rho_w f_b}{\rho_i - \rho_w} \tag{4}$$

This is the analytical basis for formulating target proxy data with known variables. The resulting simplified proxy model used is:

$$h_i \sim \frac{\rho_w C + \alpha \rho_s}{\rho_w - \rho_i} \tag{5}$$

This equation transforms a complex, coupled physical system into a simplified, mathematically tractable target proxy for our neural network.

### 4.1.1 CONCEPTUAL JUSTIFICATION FOR THE PROPOSED PROXY MODEL

The specific proxy model (Equation 5) used in our implementation includes sea ice concentration ($C$) and snow albedo ($\alpha$).

- Ice thickness is physically zero when Sea Ice Concentration ($C$) is zero. Therefore, $C$ acts as a critical constraint and scaling factor. While not physically equivalent, SIC often serves as an empirical proxy or normalization term for the bulk sea ice properties in a given area. Thicker, more stable, and more continuous ice (higher $h_i$ and $f_b$) tends to be associated with higher SIC. In a simplified model, if freeboard cannot be measured, one may substitute a function of the more easily observable area coverage (SIC), especially when modeling the mean or volume over a large grid box.

- The use of snow albedo ($\alpha$) as a proxy for snow depth ($h_s$) is justified because $\alpha$ is a highly sensitive and easily observable indicator of snow cover. Only a few centimeters of snow are required to exceed the optical depth, achieving the maximum possible albedo, meaning $\alpha$ effectively signals the presence and sufficiency of the snow layer. Since albedo is critical to the ice-albedo feedback loop and is measured reliably by remote sensing, it serves as a robust and practical stand-in for the difficult-to-measure physical depth $h_s$ in large-scale sea ice models.

## 4.2 MODEL ARCHITECTURE

The PhysE-Inv framework is a recurrent sequence modeling architecture designed to address the ill-posed nature of physical inverse problems under data scarcity. As depicted in Figure 1, our methodology achieves this by integrating three novel conceptual components: Surjective Inverse Mapping, Physics Encoding, and Physics-Guided Contrastive Learning, which together guide the model toward physically consistent solutions.

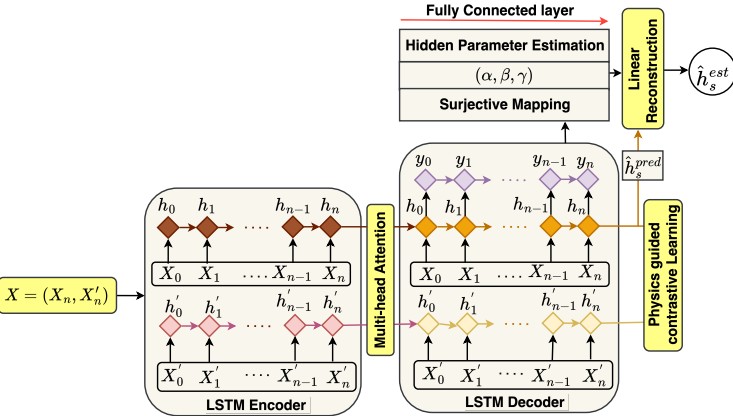

Figure 1: The Physics-Encoded Inverse (PhysE-Inv) framework outlines the fusion of physical constraints, representation learning, and a surjective inverse mapping approach based on the hydrostatic balance proxy. $T$ represents the final observation time in the input sequence.

**Sequence Latent Representation ($\mathbf{z}_T$):** The foundational step is the reliable transformation of the observable input time series $\mathbf{X} = \{x_1, \ldots, x_T\}$ (where $T = 10$ days) into a robust, contextually aware latent state ($\mathbf{z}_T$). This latent vector serves as the complete representation of the input history for all subsequent inverse operations.

**Encoder-Decoder Architecture:** We employ a standard LSTM encoder-decoder structure, leveraging the LSTM's efficiency in modeling the long-range temporal dependencies crucial for climate memory effects. The encoder sequentially processes the input, accumulating history in its states:

$$h_t^{\text{enc}}, c_t^{\text{enc}} = \text{LSTM}^{\text{enc}}(x_t, h_{t-1}^{\text{enc}}, c_{t-1}^{\text{enc}})$$

**Multi-Head Self-Attention:** A multi-head self-attention mechanism refines the encoder outputs, specializing in identifying and weighing non-local temporal dependencies that influence regional physics. The attention-enhanced input $a_t$ is computed as:

$$a_t = \text{Attention}(q_t, K, V) = \sum_{j=1}^{T} \alpha_{tj} v_j, \quad \text{where } \alpha_{tj} = \frac{\exp\left(\frac{q_t^\top k_j}{\sqrt{d_k}}\right)}{\sum_{l=1}^{T} \exp\left(\frac{q_t^\top k_l}{\sqrt{d_k}}\right)}$$

The Decoder LSTM processes this refined sequence, yielding the final hidden state, $\mathbf{z}_T = h_T^{\text{dec}}$, which is the definitive input to the inverse module.

**Surjective Inverse Mapping and Physics Encoding:** The challenge of retrieving sea ice parameters from uncertain satellite data is fundamentally an ill-posed inverse problem. To address the non-uniqueness inherent in this problem, the core novelty of our approach is the definition of a surjective inverse mapping that links the model's latent state to the final physical parameter based prediction, ensuring physical constraints are enforced dynamically. We implement the surjective inverse mapping using an MLP-based neural operator that predicts the hidden sea ice parameters $\boldsymbol{\Theta}_t = [\boldsymbol{\alpha}, \boldsymbol{\beta}, \boldsymbol{\gamma}]$ from the observed inputs. This mapping connects the abstract latent state of the model to the required physical quantities.

Specifically, the network outputs a set of raw, unconstrained predictions, which are then transformed to enforce physically meaningful constraints. Conceptually, this operation can be written as:

$$\boldsymbol{\Theta}_t = \mathrm{Transform}(\mathrm{MLP}(\mathbf{z}_T))$$

In our implementation, the latent state $\mathbf{z}_T$ (the last hidden state from the decoder) is passed through a fully connected layer to produce the raw parameters $\boldsymbol{\alpha}_{\mathrm{raw}}, \boldsymbol{\beta}_{\mathrm{raw}}, \boldsymbol{\gamma}_{\mathrm{raw}}$. These are then transformed to ensure adherence to their appropriate physical domains:

$$\boldsymbol{\alpha} = \mathrm{sigmoid}(\boldsymbol{\alpha}_{\mathrm{raw}}) \times 2 - 1$$

$$\boldsymbol{\beta} = \exp(\boldsymbol{\beta}_{\mathrm{raw}})$$

$$\boldsymbol{\gamma} = \tanh(\boldsymbol{\gamma}_{\mathrm{raw}}) \times 10$$

The raw outputs are transformed to enforce physically meaningful constraints: $\boldsymbol{\alpha}$ is passed through a scaled sigmoid to lie in $[-1, 1]$, $\boldsymbol{\beta}$ is exponentiated to ensure positivity, and $\boldsymbol{\gamma}$ is passed through a scaled tanh to lie in $[-10, 10]$. This procedure allows the network to recover dynamically varying hidden parameters in a way that respects their underlying physical constraints while naturally realizing the many-to-one (surjective) mapping from observations to the required latent physical variables.

**Physics Encoding and Reconstruction Proxy:** The estimated parameters $\boldsymbol{\Theta}_t$ are used to enforce the hydrostatic balance via a physics encoding layer. The final physics-constrained prediction, $\hat{h}_{s,t}^{\mathrm{est}}$, is defined by the following reconstruction proxy:

$$\hat{h}_{s,t}^{\mathrm{est}} = \boldsymbol{\alpha}_t \cdot \overline{\hat{h}}_{s,t}^{\mathrm{pred}} + \boldsymbol{\beta}_t \cdot \hat{h}_{s,t}^{\mathrm{pred}} + \boldsymbol{\gamma}_t \tag{6}$$

where $\hat{h}_{s,t}^{\mathrm{pred}}$ is the direct prediction, and $\overline{\hat{h}}_{s,t}^{\mathrm{pred}}$ is the mean of the intermediate predictions over the sequence $T$. The corresponding loss, $\mathcal{L}_{\mathrm{PE\text{-}pred}}$, ensures the final output respects the physics:

$$\mathcal{L}_{\mathrm{PE\text{-}pred}} = \frac{1}{T} \sum_{t=1}^{T} (\hat{h}_{s,t}^{\mathrm{pred}} - \hat{h}_{s,t}^{\mathrm{est}})^2 \tag{7}$$

**Physics-Guided Contrastive Learning (PGCL)** The latent space is guided by the physical principle of **invariance**. Because Gaussian noise augmentation ($\mathbf{X}'$) does not alter the underlying physical characteristics of the original input ($\mathbf{X}$), the pair $(\mathbf{X}, \mathbf{X}')$ represents two physically equivalent observations. The model is therefore encouraged to produce embeddings that remain stable under observational noise. To enforce this invariance, we adopt a contrastive learning objective inspired by the Normalized Temperature-Scaled Cross-Entropy (NT-Xent) loss. For each batch of $N$ sequences, we construct $N$ positive pairs, resulting in a $2N$-sample contrastive batch. Given a pair of embeddings $(\mathbf{z}_i, \mathbf{z}_j)$, the loss promotes high similarity for the positive pair while pushing apart all other embeddings in the batch:

$$\mathcal{L}_{\mathrm{Contrastive}} = -\frac{1}{2N} \sum_{i=1}^{2N} \log \left( \frac{\exp(s(\mathbf{z}_i, \mathbf{z}_{j(i)})/\tau)}{\sum_{k \neq i} \exp(s(\mathbf{z}_i, \mathbf{z}_k)/\tau)} \right) \tag{8}$$

where $s(\cdot, \cdot)$ denotes cosine similarity and $\tau$ is the temperature.

**Contrastive Objective Simplification:** Our contrastive regularizer is not equivalent to NT-Xent, nor is it intended to replicate its temperature-scaled formulation. Instead, we adopt a simplified objective that captures only the aspect relevant to our setting: encouraging separation between physically inconsistent parameter reconstructions while keeping consistent samples close in the latent space. This lightweight formulation avoids the heuristic temperature tuning and large-batch dependence of standard contrastive losses, which are unnecessary for our small, structured physical parameter space. The goal is therefore not to implement a canonical contrastive learning loss, but to introduce a minimal stability-inducing term that supports the surjective, invertibility-aware architecture.

## 5 BASELINES

Our study's primary objective is conceptual: to rigorously evaluate how enforcing a surjective, invertibility-aware mapping enables the stable recovery of latent physical parameters under data scarcity. While acknowledging the presence of general state-of-the-art architectures in time-series forecasting, our baseline selection is deliberately focused on controlled, ablative comparisons to isolate the specific impact of the PhysE-Inv framework's conceptual innovations.

Table 1: Comparison of model predictions with and without parameter estimation (PE).

| Model | Without PE | | With PE | |
|---|---|---|---|---|
| | **MSE** | **RMSE** | **MSE** | **RMSE** |
| LSTM | 0.4679 | 0.6840 | 0.4545 | 0.6742 |
| NeuralODE | 0.5066 | 0.7117 | 0.4926 | 0.7018 |
| ResNet50 | 0.4308 | 0.6563 | 0.4315 | 0.6569 |
| BiLSTM | 0.5263 | 0.7255 | 0.5177 | 0.7195 |
| **PhysE-Inv** | **0.3942** | **0.6278** | **0.3568** | **0.5973** |

In domains like physics-guided inverse modeling, many architectures in the literature are custom-built to solve tightly scoped problems with specific assumptions and non-linear partial differential equations, often lacking the generalizability needed for broader comparisons. They frequently only present sample-based ablations but not performance comparisons between state-of-the-art architectures Raissi et al. (2019); Ghosh et al. (2022). For example, we fundamentally differ from PINNs Raissi et al. (2019), which use automatic differentiation to solve PDEs for numerical simulation. Therefore, in our study, to facilitate a fair comparison across different architectural paradigms, we chose to augment each of our selected baselines with an inverse modeling module, enabling them to perform the same joint parameter estimation and prediction task as our proposed PhysE-Inv framework. Furthermore, we acknowledge the existing gap in the literature regarding direct comparisons between general-purpose architectures adapted for inverse modeling and those models specifically designed for particular inverse problems. It is noteworthy that our proposed model combines inverse modeling (specifically through its parameter estimation process) with physics-guided contrastive learning (Fig. 1).

To ensure a fair analysis, all baselines are capacity-matched to our PhysE-Inv model. Furthermore, where applicable, they incorporate the same invertibility adjustment to ensure performance differences are attributable to our novel components (Physics Encoding and Physics-Guided Contrastive Learning), not differences in overall model scale. Our choices represent conceptually distinct classes of time-series modeling with noisy, sparse real-world data:

- LSTM serves as the minimal recurrent baseline, directly aligning with the core temporal backbone of our encoder-decoder structure. This comparison quantifies the added value of our physics constraints over a standard sequence modeling approach.

- BiLSTM is included to evaluate whether performance gains are due to the structural enforcement of physical rules, or simply from leveraging broader (non-causal) temporal context.

- Neural ODE offers a direct comparison to an intrinsically continuous-time modeling approach, often considered ideal for latent physical dynamics, testing the efficacy of our explicit physics-guided architecture.

- ResNet-50 (1D variant) provides a non-recurrent, deep convolutional benchmark. This tests whether the temporal memory inherent in the LSTM is necessary when compared to a capacity-matched architecture that models implicit, discretized dynamics through hierarchical feature extraction.

These models collectively form a set of conceptually aligned baselines that effectively isolate and benchmark the impact of our proposed invertibility principles in estimating time-varying physical parameters.

## 5.1 RESULTS AND DISCUSSION

Table 1 presents a comparison of the predictive performance of four baseline models used in this study with the proposed inverse model PhysE-Inv. The baseline models are LSTM, NeuralODE, ResNet50, and BiLSTM. The evaluation considers their performance under two different model settings: the first setting, referred to as the base, reflects prediction without the incorporation of hidden characteristics that arise from parameter estimation and inverse mapping, while the other employs parameter estimation and thus includes these hidden characteristics. Model efficacy is quantified using two standard error metrics (MSE and RMSE). Lower values for both metrics indicate a higher degree of predictive accuracy.

Table 2: Ablation study comparing PhysE-Inv performance with and without supervised contrastive learning (SCL)

| Training Data | Without SCL | | With SCL | |
|---|---|---|---|---|
| | MSE | RMSE | MSE | RMSE |
| **Sample 1** (80%) | 0.6601 | 0.8125 | **0.5926** | **0.7698** |
| **Sample 2** (60%) | 0.6588 | 0.8117 | **0.6037** | **0.7770** |
| **Sample 3** (50%) | 0.8675 | 0.9314 | **0.7940** | **0.8911** |

The results indicate that the application of parameter estimation generally correlates with a reduction in both MSE and RMSE across the evaluated models, suggesting an enhancement in predictive accuracy. Notably, the proposed model demonstrates the lowest error values both in its base form (without inverse modeling and parameter estimation) and in its form with inverse modeling and parameter estimation (PhysE-Inv), indicating superior performance in this specific prediction task. While parameter estimation shows improvements for other models, the extent of this improvement varies. For instance, both LSTM and NeuralODE exhibit a substantial decrease in error, whereas the performance of ResNet50 remains relatively consistent. This suggests that the effectiveness of parameter estimation in improving predictive accuracy is model-dependent. Additionally, our proposed model consistently shows better performance across all comparisons.

We conducted ablation experiments with and without contrastive learning (CL), which typically means the inclusion or exclusion of contrastive loss. The table (2) presents the results of ablation experiments conducted to examine the effectiveness of CL in our proposed model, using three different training data samples. Following Ghosh et al. (2022), we performed an ablation study by reducing the training data sample size to evaluate the impact of CL. Specifically, Sample 1 used 80% of the training data, Sample 2 used 60%, and Sample 3 used 50%. Our experimental results indicate that the model trained with CL consistently outperformed the model trained without CL across all three sample sizes, yielding lower MSE and RMSE values, which generally suggest more accurate predictions. It is important to note that an essential part of our model architecture is the inclusion of CL in addition to the core approach of parameter estimation for physics-based components. Therefore, the enhanced performance observed in the ablation study may result from CL's ability to learn better representations, from the improved

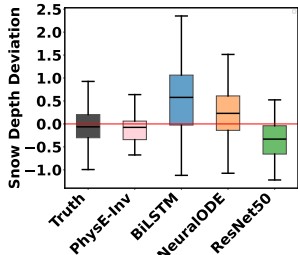

Figure 2: Comparison of model performance: box plot of snow depth deviations.

the ablation study may result from CL's ability to learn better representations, from the improved

reconstruction of physical relationships via parameter estimation, or from both. Since 'Physics Encoding' (surjective inversion with parameter estimation) plays a key role in embedding physical knowledge into the model, removing it entirely would fundamentally alter and misrepresent our proposed model's structure. Therefore, it doesn't make sense to do an ablation study without the physics loss, as such a model would no longer represent our core approach.

The box-and-whisker plots in Figure 2 illustrate the distributions of snow depth anomalies for the ground truth, baseline models, and the proposed PhysE-Inv. A horizontal dashed red line at zero represents perfect alignment with the average snow depth condition. Among the baselines, BiLSTM shows reasonably consistent performance, with a median close to zero and a spread comparable to the ground truth, though it produces fewer negative outliers. This suggests that BiLSTM captures the overall shape of the distribution while underrepresenting extreme values. ResNet50 also has a median near zero, but its predictions show a narrower spread than the ground truth, indicating the model may underestimate the full variability of the anomalies. In contrast, NeuralODE exhibits greater variability with a median slightly above zero, suggesting an upward bias, and contains more significant negative outliers, pointing to reduced stability and increased deviation from the true distribution. PhysE-Inv demonstrates the closest agreement with the ground truth. Its predictions have medians very near zero and a spread that matches the true anomaly distribution. The relatively low number of outliers indicates stable predictions and an accurate representation of the overall distribution. Overall, PhysE-Inv provides the most reliable and consistent estimates, successfully capturing both the central tendency and variability of snow depth anomalies.

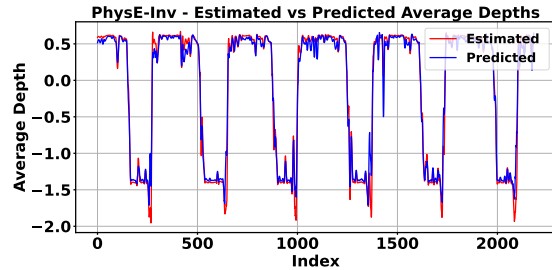

Figure 3: Time series of predicted and estimated mean snow depth seasonal pattern

Figure 3 illustrates the strong predictive performance of the proposed PhysE-Inv model on the test dataset, validated by the close alignment between the predicted and estimated normalized snow depth proxy time series. The model effectively captures the highly dynamic and cyclical behavior of the proxy, demonstrating exceptional fidelity in tracking both the long-term seasonal variation and the abrupt transitions that define extreme events. Specifically, the model accurately follows the sharp drops from the high baseline (normalized value near 0) down to the seasonal minimums (near -2), which likely correspond to critical physical processes such as rapid compaction or melt events. While minor predictive errors, characterized by slight smoothing or phase lag, are primarily localized around these abrupt transitions, the overall temporal fidelity and quantitative accuracy remain high. This close correspondence confirms the model's robustness and its ability to generalize the complex non-linear relationship between the input snow density field and the normalized snow depth proxy.

## 6 CONCLUSION AND FUTURE WORK

We presented PhysE-Inv, a physics-encoded inverse modeling framework designed to estimate time-varying hidden geophysical parameters from sparse observations to predict real-world Arctic snow depth. The core novelty lies in combining a lightweight inverse head with physics-guided contrastive learning to achieve a surjective inversion strategy . PhysE-Inv accurately reconstructed the seasonal evolution of snow depth and consistently outperformed both physics-regularized and data-driven baselines, demonstrating that simple, interpretable physics encoding yields substantial gains over generic PIML methods when data are limited. This approach establishes a robust methodology for handling ill-posed inverse problems in environmental science, paving the way for more reliable estimation of unobserved variables globally. Future work will incorporate Bayesian uncertainty quantification to validate learned parameters against independent remote-sensing and in-situ sources.

### AUTHOR CONTRIBUTIONS

This work uses a large language model to refine and clarify the language, improve sentence structure, and polish the writing.

### ACKNOWLEDGMENTS

Use unnumbered third level headings for the acknowledgments. All acknowledgments, including those to funding agencies, go at the end of the paper.

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

# A APPENDIX

## A.1 REGION OF IMPORTANCE

The behavior of sea ice and its overlying snow is analyzed within a specific geographical domain. This Region of Importance, the central Arctic Ocean, is an essential component of the global climate system. Figure 4 illustrates the exact spatial extent of this region, which is used for extracting all atmospheric and oceanographic parameters. To extract spatially aggregated information for our time series prediction, spatial averaging was applied across the grid points within the orange highlighted area to produce daily time series. Z-score normalization was then performed using the training set.

Figure 4: Map showing the spatial extent of the central Arctic Ocean, highlighting (orange line) the region used for data collection.

## A.2 Model Parameters and Notation

This appendix provides a detailed reference for all physical parameters and variables discussed in the modeling chapters. Table 3 defines the essential atmosphere and ocean parameters, their respective notations, and whether they are treated as constant values or time series.

Table 3: Atmosphere and Ocean Parameters (Features)

| Category | Parameter | Notation | Type |
|---|---|---|---|
| Known parameters | Snow albedo | $\Omega$ | Time series |
| | Snow density | $\rho_{\mathrm{s}}$ | Time series |
| | Sea ice concentration | $C$ | Time series |
| | Seawater density | $\rho_{\mathrm{w}}$ | Constant |
| | Sea ice density | $\rho_{\mathrm{i}}$ | Constant |
| Unknown parameters | Ice thickness | $h_{\mathrm{i}}$ | Time series |
| | Ice freeboard | $f_{\mathrm{b}}$ | Time series |
| | Snow depth | $h_{\mathrm{s}}$ | Time series |

## A.3 Non-Uniqueness and the Ill-Posed Problem

The conceptual framework illustrates why retrieving sea ice thickness ($h_{\mathrm{i}}$) or snow depth ($h_{\mathrm{s}}$) from remote-sensing measurements constitutes an ill-posed inverse problem. This is demonstrated by contrasting the stable forward mapping with the non-unique inverse mapping implied by the principle of hydrostatic balance.

**Forward Problem (Well-Posed):**
$$(h_{\mathrm{s}},\ f_{\mathrm{b}}) \ \longrightarrow \ h_{\mathrm{i}}$$
Given the true snow depth $h_{\mathrm{s}}$ and freeboard $f_{\mathrm{b}}$, the hydrostatic equation produces a unique ice thickness $h_{\mathrm{i}}$. This mapping is stable and well-defined.

**Inverse Problem (Ill-Posed):**
$$h_{\mathrm{i}} \ \longrightarrow \ (h_{\mathrm{s}},\ f_{\mathrm{b}})$$
Given only the ice thickness $h_{\mathrm{i}}$, there exist infinitely many pairs $(h_{\mathrm{s}},\ f_{\mathrm{b}})$ that satisfy the hydrostatic balance. Thus, the inverse mapping is non-unique, unstable, and therefore ill-posed.

**Practical Implications:** In satellite altimetry, the inputs to the forward computation, especially snow depth, are highly uncertain. Prior studies show that uncertainty in snow depth is the dominant source of error in satellite-derived sea ice thickness estimates (Kwok & Cunningham, 2008). Consequently, small perturbations in $h_{\mathrm{s}}$ or $f_{\mathrm{b}}$ can produce large variations in the inferred thickness $h_{\mathrm{i}}$.

**Implication for Learning-Based Methods:** Although the model predicts $h_{\mathrm{i}}$ rather than the inverse quantities, the ambiguity inherent in the measurement space implies that multiple noisy or uncertain input configurations can map to similar thickness values. This structural non-uniqueness contributes to the ill-posedness of the learning problem and motivates incorporating physics-guided constraints or regularization.

## A.4 Conceptual Justification: Why Surjectivity is Necessary

The fundamental difficulty in retrieving sea ice parameters stems from the nature of the mapping between the observed variables and the target quantities. Specifically, the relationship between inputs and the output is not a simple one-to-one correspondence.

Figure 5: (a) Bijective mapping: unique input-output relationship. (b) Surjective mapping: multiple inputs to one output.

The necessary mathematical framework is best understood by contrasting two types of mappings (Figure 1):

**Bijective Mapping (Hypothetical):** This mapping assumes the inverse problem is fully reversible (one-to-one). While mathematically ideal for unique solutions, this model does not reflect the inherent ambiguity and coupled dependencies present in real-world sea ice measurements.

**Surjective Mapping (The Real-World Model):** This mapping implies that the inverse problem is not uniquely reversible (many-to-one). This structure correctly models the non-uniqueness where multiple input measurement pairs can contribute to a single output value (e.g., the same ice thickness).

The surjective assumption is therefore necessary because it acknowledges that every physically possible thickness value has at least one corresponding input measurement pair. By adopting this many-to-one structure, the model effectively guarantees that a solution exists in the output parameter space for every input latent state $\mathbf{z}_T$, overcoming the rigid, ill-posed requirements of traditional inverse modeling techniques.

## A.5 ADDITIONAL RESULTS

Figure 6 compares the Probability Density Function (PDF) of the ground truth and the PhysE-Inv model's predictions. This comparison provides a more comprehensive evaluation than point-estimate metrics like MSE by assessing the model's ability to learn the underlying statistical distribution of the data. In climate science, where a perfect one-to-one match in geospatial grids is often not expected, the alignment of the probability distributions becomes a more crucial evaluation criterion. A strong correspondence between the predicted and true PDFs indicates that the model is not merely a regression function for individual data points but is capable of generalizing the data's generative process. This is crucial for capturing the system's overall statistical behavior, including the frequency and likelihood of different outcome magnitudes.

The PhysE-Inv model successfully captures the unimodal nature of the measured deviations in the ground truth distribution (Fig. 6). By incorporating physics encoding, it captures real-world variabilities, reducing prediction error and producing more physically consistent and robust predictions.

## A.6 IMPLEMENTATION

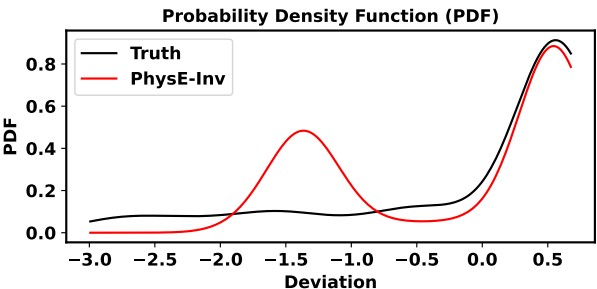

Figure 6: PDF of mean snow depth anomalies for true values and PhysE-Inv predictions.

Table 4: Model Training and Architectural Hyperparameters

| Category | Details |
|---|---|
| ***Data and Setup*** | |
| Input Feature | Snow density field (time series) |
| Target | Normalized snow depth proxy |
| **Input/Target Shapes** | **[Batch size, Sequence Length, Features]** |
| Train $X/Y$ Shape | $[8757, 10, 1]$ |
| Test $X/Y$ Shape | $[2183, 10, 1]$ |
| Supervision Strategy | Prediction at the final time step ($t = 10$) |
| ***Optimization*** | |
| Loss Function | Mean Squared Error (MSE) |
| Optimizer | Adam, $\eta = 0.0005$ |
| Batch Size | 16 |
| Training Epochs | 500 |
| Implementation / Hardware | PyTorch on NVIDIA V100 GPU |
| ***Architecture*** | |
| Encoder/Decoder | 2-layer LSTM (64 hidden units) |
| LSTM Dropout Rate | 0.4 |
| Attention Mechanism | 4-headed self-attention |
| Prediction Head | Fully connected layer |
| Physical Parameter Head | 3-layer FFN with ReLU |
| Output Transformation | Ensures stable parameter ranges |