# OpenReview forum: "PhysERL-Inv: A Physics-Encoded Inverse Modeling Approach for Arctic Snow Depth Prediction"
_ICLR.cc/2026/Conference — Submitted to ICLR 2026_

### Official Review · Reviewer_t4ue · 2025-10-30

**Soundness:** 2
**Presentation:** 2
**Contribution:** 1
**Rating:** 2
**Confidence:** 4

**Summary:**

The paper proposes a physics-encoded inverse modeling approach to estimate Arctic snow depth from reanalysis inputs. A sequence model (RNN/attention + MLP head) is coupled to a linearized hydrostatic relation and regularized with a contrastive term built from noise-perturbed views of the inputs. On a central-Arctic, daily ERA-style dataset, the method beats several neural baselines and tracks seasonal dynamics.

**Strengths:**

1. **Physics-guided objective:** Encoding a (linearized) hydrostatic relation into the learning target promotes physically plausible estimates rather than unconstrained regression.

2. **Signal in data-scarce regimes:** The contrastive regularizer appears to stabilize training when labels/coverage are limited.

**Weaknesses:**

1. **Foundation-model baselines for Earth/physics tasks:** There’s no comparison to ClimaX [1] (a pre-trained foundation model for weather/climate that adapts to heterogeneous variables and scales), which already shows strong transfer on ERA/CMIP tasks after lightweight fine-tuning. Even a small ClimaX adapter fine-tuned to snow-relevant channels would be an informative baseline, or a complementary pretrain-then-physics-tune strategy.

2. **External truth & circularity.** The target appears (from the description) to be derived from the same hydrostatic relation the model encodes. That risks **learning to the proxy**, which basically means evaluating the model on the same training set. Independent validation against satellite/freeboard-derived snow products, OIB tracks, or buoy stations is critical to establish real-world value, not just self-consistency.

3. **Physics-informed peers beyond vanilla nets.** The baselines don’t include modern physics-aware methods (PINNs [3], neural operators with constraints, hybrid data-assimilation losses). Given ClimaX [1] and PhysiX [2] show strong data-driven physics fidelity at scale, this weakens the claims of state-of-the-art for this work.

4. **Supervised contrastive clarity.** Using noise-perturbed views as positives is data-augmentation/self-supervised flavored; it would help to justify the “supervised” label and analyze sensitivity to temperature, batch size, and negative sampling.


[1] Nguyen, Tung, et al. "Climax: A foundation model for weather and climate." arXiv preprint arXiv:2301.10343 (2023).

[2] Nguyen, Tung, et al. "PhysiX: A Foundation Model for Physics Simulations." arXiv preprint arXiv:2506.17774 (2025).

[3] Raissi, Maziar, Paris Perdikaris, and George E. Karniadakis. "Physics-informed neural networks: A deep learning framework for solving forward and inverse problems involving nonlinear partial differential equations." Journal of Computational physics 378 (2019): 686-707.

**Questions:**

1. Please add baselines from (i) PINNs/physics-loss regularized nets; (ii) a simple hydrostatic-only regression. This will help isolate the gain from your inverse head vs. generic physics regularization.

2. For ablations, please compare between removing each loss term (prediction, physics, contrastive) and each architectural block (attention, inverse-param head).

3. What independent observations of snow depth (e.g., ICESat/ICESat-2 freeboard-derived products, Operation IceBridge, in-situ buoys) can you use to validate the learned model and rule out circularity with Eq. (1)? If none, please justify why the proxy target suffices. Else, please train on one region/period, test OOD on another.

4. Please precisely define the target space and explain how your architecture or loss enforces a surjective inverse (or provide a counterexample/limitation). Any theoretical or empirical evidence (e.g., coverage metrics over plausible parameter sets)?

5. Clarify why calling the objective “supervised” is appropriate when positives are noise-perturbed views; discuss sensitivity to $\tau$, batch size, and negative sampling strategy.

---

### Official Review · Reviewer_GL7T · 2025-10-31

**Soundness:** 2
**Presentation:** 2
**Contribution:** 2
**Rating:** 2
**Confidence:** 3

**Summary:**

This paper proposes PhysERL‑Inv, a hybrid framework that combines (i) an LSTM–attention autoencoder, (ii) an “inverse” module that estimates three scalars and uses a simplified linear relation to compute snow depth, and (iii) a contrastive objective.

The physics component is motivated by the hydrostatic balance, and training is performed on ERA5 after spatial averaging over a central‑Arctic box. Targets are proxy labels derived from the hydrostatic relation; evaluation compares to several deep‑learning baselines (LSTM, BiLSTM, Neural ODE, ResNet‑50). The authors report lower MSE/RMSE than baselines.

**Strengths:**

1. Timely topic & societal relevance. Snow on sea ice is a key uncertainty for altimetry‑based sea‑ice thickness retrievals and Arctic mass‑balance studies, reducing its error has direct scientific value.
2. Physics‑guided ML framing. The paper aims to embed a first‑principles relation (hydrostatic balance) into a neural architecture, aligning with an active research direction on integrating scientific knowledge with machine learning. Surveys and tutorials motivate this paradigm and its benefits for data‑sparse scientific regimes.
3. Clear architecture & training description. The manuscript describes the encoder–attention–decoder stack, the inverse parameter head, and the composite loss; training details (optimizer, sequence length, batch size, chronology split) are provided.

**Weaknesses:**

0. The submission used a wrong template for ICLR 2025.

1. Baselines and claims are inconsistent or mismatched:
ResNet‑50 is a 2D image backbone; using it as a time‑series baseline is questionable without a rigorous adaptation (e.g., TCNs, Transformers for sequences) and hyperparameter parity. More appropriate baselines include Temporal Convolutional Networks, sequence Transformers or other types on physics informed architectures.
The text claims a “32.20% reduction vs the best‑performing baseline (BiLSTM),” but BiLSTM is worst among listed baselines (highest MSE/RMSE). This undermines the quantitative narrative.

2. Data pipeline and reproducibility gaps: Inputs listed are snow albedo, snow density, and sea‑ice concentration from ERA5, spatially averaged over a region; yet the hydrostatic mapping requires freeboard and ice thickness. The manuscript says these variables are “processed in the context of” thickness/freeboard but provides no retrieval method, and no uncertainty treatment for them during label generation.

3. Clarity: “During the winter season, both true and predicted snow depths show a downward trend, indicating a period of accumulation.” It should be 'upward'.

4. It is unclear about how to get the 'standard deviation' in Fig3. If it is part of the contribution, then the uncertainty training methods should be compared.

**Questions:**

Please refer to the weakness.

---

### Official Review · Reviewer_Vw7G · 2025-11-01

**Soundness:** 2
**Presentation:** 3
**Contribution:** 2
**Rating:** 2
**Confidence:** 3

**Summary:**

PhysERL-Inv is a hybrid physics-encoded inverse modeling method to predict Arctic snow depth. The authors encode a hydrostatic balance equation into a sequence model, use supervised contrastive representation learning to shape latent representations, and invert the physics-encoded mapping to estimate hidden parameters that improve snow-depth prediction. They report substantial error reductions versus several neural baselines (reporting ≈20% improvement overall).

**Strengths:**

- Clear applied motivation. Snow depth over Arctic sea ice is an important but sparsely observed variable — the problem and practical need are well framed.

- Nice hybrid approach. Combining an explicit physics relation (hydrostatic balance) with representation learning and an inverse mapping is conceptually sensible for data-sparse geoscience tasks. The architecture and training pipeline (encoder–decoder LSTM + self-attention, contrastive loss + MSE) are described and reasonable.

- Solid empirical improvements & ablations. The paper compares to a set of neural baselines (LSTM, BiLSTM, NeuralODE, ResNet50) and shows quantitative wins (PhysERL-Inv MSE 0.3568 vs others; the table shows 17–32% relative changes on MSE/RMSE) and an ablation showing supervised contrastive learning (SCL) helps in low-data regimes. Those results are presented clearly.

**Weaknesses:**

- Limited methodological novelty and analysis. The paper mainly integrates existing components — LSTM-based sequence modeling, supervised contrastive learning, and physics encoding via hydrostatic balance — without introducing a new learning algorithm or architectural mechanism. While the integration is well-motivated, the paper lacks deeper analysis or theoretical insight into why the combination works or how the proposed “surjective inversion” differs fundamentally from standard inverse modeling or PINN-style methods.

- Baseline coverage is incomplete. The study compares primarily against conventional neural baselines (LSTM, BiLSTM, NeuralODE, ResNet50), but omits stronger physics-informed or operator-learning approaches (e.g., PINNs, DeepONet, FNO). Including or at least discussing these would be necessary to convincingly demonstrate advantages at the level expected for top ML venues.

- Limited scope and generalization evaluation. Experiments are restricted to one Arctic dataset and temporal range, with no tests on cross-region or out-of-distribution generalization. Broader validation across different spatial or temporal domains would help establish robustness and strengthen claims of general applicability.

- Formatting issue. The submission uses the 2025 ICLR template instead of the required 2026 version, which should be corrected for compliance.

**Questions:**

Could you add ablations for (a) physics encoding strength (e.g., ablating terms in the hydrostatic relation), (b) contrastive loss weight, and (c) latent-dimension size.

---

### Official Review · Reviewer_mdV3 · 2025-11-01

**Soundness:** 2
**Presentation:** 2
**Contribution:** 3
**Rating:** 4
**Confidence:** 3

**Summary:**

This paper proposed a physics-encoded representation learning method for the prediction of Arctic snow depth. The proposed method consists of an encoder-decoder based LSTM with multi-head self-attention mechanism to capture temporal dependencies, an MLP-based inverse modeling module to encode physics knowledge of the hydrostatic balance equation, and a supervised contrastive learning objective for augmentation. Experimental results showed that the proposed method outperformed the baselines, and the ablation study further highlighted the importance of supervised contrastive learning.

**Strengths:**

1. The paper is well-organized.
2. The experiments are conducted on the real-world dataset, making the results more convincing and demonstrating the potential of the proposed method to real-world applications.
3. The proposed method yielded superior performance compared to all the baselines.

**Weaknesses:**

1. The inverse modeling of the proposed method leverages the physical relationship in the hydrostatic balance equation. However, it is unclear how the original equation in Equation (1) is simplified to Equation (3).
2. The ablations regarding the contributions of the multi-head self-attention mechanism and the physics encoded loss are missing. The latter one is especially important for the proposed method.
3. The conclusion section is missing.

**Questions:**

1. According to the authors, the output of the inverse modeling $\hat{s}_t$ is predicted snow depth, and the output of the LSTM $\hat{y}_t$ is also predicted snow depth. Are they equivalent? If so , the Equation (3) can be learned as an identical function where $h^{\text{eq}}_f=0$, $h_i=0$ and $\rho_i=0$. How to avoid this situation?
2. What exactly are queries, keys and values for the multi-head self-attention blocks? Are they derived from hidden states? If so, how does it handle the sequences with various lengths?
3. The description of supervised contrastive learning is confusing. $X’$ has different definitions: a physically-similar sample and an augmented sequence of $X$. From my understanding, it can be both. The augmented sequence of $X$ serves as the positive sample, while the physically-similar sample serves as the negative sample. However, I’m not sure if the perturbation is also added to the physically-similar sample.
4. [Minor] In Section 3.2 (Encoder-LSTM), “the LSTM uses the current input $x_t$ and its internal memory to calculate the hidden state $h_{t−1}$." Should it be $h_t$?
5. [Minor] In Figure 1, the inputs to the LSTM decoder are all augmented sequence. Should it be $a_t$ and its augmented version instead?

---

### Meta-Review · Area_Chair_SJap · 2025-12-08

**Summary:**

The reviewers raised multiple concerns including limited novelty, inconsistent claims/baselines, and limitations in the experiments.  The responses go some way to addressing some of these, but my best judgment is that it is quite unlikely that most (if any) of the reviewers would have been convinced enough to recommend acceptance.  This is due to several factors including brief responses, responses that acknowledge the limitation raised, and acknowledgments of errors/omissions in the original version.

**Reviewer Concerns:**

It is difficult to list these one by one, but just as one example, the following was quite a serious concern but with not much said in response: "Baselines and Claims are Inconsistent or Mismatched: We clarify that our ResNet-50 baseline uses a rigorous 1D Convolutional adaptation specifically designed for time-series, which is a common practice in capacity-matched benchmarking."

In addition, several of what the reviewers found lacking was relegated to future work, e.g., "We agree that broader spatiotemporal validation is important, and we would like to address this in future work."

**Reviewer Scores:**

It's possible that some of the scores might increase, but very likely that most would remain below the acceptance threshold, due to the low starting point (4/2/2/2) and the points mentioned above.

---

### Decision · Program_Chairs · 2026-01-26

Reject